# Management and Outcomes of Older Patients (Age ≥ 70 Years) with Advanced Soft Tissue Sarcoma and Role of Geriatric Assessment and Oncological Multidimensional Prognostic Index (Onco-MPI) in a Real-World Setting

**DOI:** 10.3390/cancers15041043

**Published:** 2023-02-07

**Authors:** Benedetta Chiusole, Ilaria Tortorelli, Antonella Galiano, Fabio Murtas, Selma Ahcene-Djaballah, Giuseppina Tierno, Eleonora Bergo, Alberto Banzato, Maura Gatti, Antonio Di Maggio, Giuseppe Sergi, Marco Rastrelli, Marta Sbaraglia, Vittorina Zagonel, Antonella Brunello

**Affiliations:** 1Oncology 1, Department of Oncology, Istituto Oncologico Veneto IOV-IRCCS, 35128 Padova, Italy; 2Department of Surgery, Oncology and Gastroenterology (DISCOG), University of Padua, 35128 Padua, Italy; 3Oncology 3, Department of Oncology, Istituto Oncologico Veneto IOV-IRCCS, 35128 Padova, Italy; 4Cardiology, Istituto Oncologico Veneto IOV-IRCCS, 35128 Padova, Italy; 5Oncologic Radiology Unit, Department of Radiology and Medical Physics, Istituto Oncologico Veneto IOV-IRCCS, 35128 Padova, Italy; 6Department of Medicine (DIMED), Geriatric Division, University of Padova, 35122 Padua, Italy; 7Soft-Tissue, Peritoneum and Melanoma Surgical Oncology Unit, Istituto Oncologico Veneto IOV-IRCCS, 35128 Padua, Italy; 8Surgical Pathology Unit, Padua University Hospital, 35121 Padua, Italy

**Keywords:** onco-MPI, CGA, geriatric assessment, elderly, sarcoma, chemotherapy

## Abstract

**Simple Summary:**

At Istituto Oncologico Veneto we are providing a geriatric assessment to all patients aged 70 years and older since 2003. Soft tissue sarcoma are really rare neoplasm and we, as a referral centre, evaluate a high volume of patients, so we decided to conduct this study to describe the geriatric multidisciplinary management and also the role the geriatric tools in the decision making and in assessing the prognosis.

**Abstract:**

**Background:** Incidences of soft tissue sarcomas (STS) steadily increase with age. Yet, despite the high prevalence in advanced age, older patients (pts) are underrepresented in sarcoma clinical trials and evidence-based guidelines for chemotherapy are lacking. International oncological societies suggest using geriatric tools to evaluate older patients with cancer to optimise treatment indication. Comprehensive geriatric assessment (CGA) is a multidimensional assessment of older subjects, based on which pts can be classified as fit, vulnerable or frail. Onco-MPI (multidimensional prognostic index) is a CGA-based score which also considers tumour characteristics, classifying pts into three risk groups of death at one year: high-risk, intermediate-risk and low-risk. **Methods:** This is a single-centre retrospective study which aims at describing real-word management and outcomes of older pts with advanced stage STS and at assessing the ability of CGA and onco-MPI to predict survival in these pts. Consecutive pts with advanced stage STS aged 70 years or older and treated at the Istituto Oncologico Veneto from January 2009 to June 2020 were retrieved from a prospectively maintained database. Pts’ demographics, CGA assessments and tumour characteristics were analysed. Statistical analysis was performed with R version 3.4.3 **Results:** Out of 101 pts, with a median age of 77 years, 76 received chemotherapy (75.3%), which was anthracycline-based for 46 pts (60.5%). Anthracyclines were used in a higher proportion in fit pts (58.9% fit vs. 45.1% vulnerable vs. 12.5% frail pts). Frail pts and pts in the onco-MPI high-risk group experienced a higher rate of chemotherapy-related toxicities. Median OS was 13.8 months (95% CI 11.3–17.7 months). According to CGA, the median OS was 19.53 months (95% CI 15.23–36.8) for fit pts, 12.83 months (95% CI 9.7–17.5) for vulnerable and 7.75 months (95% CI 2.73–30) for frail pts (*p* = 0.005). Onco-MPI confirmed a predictive value for 1-year survival with intermediate risk pts not reaching a median OS at 1 year, and high-risk pts having a median one-year OS of 11.5 months (95%CI 9.7–NA), *p* = 0.02. In multivariate analysis, onco-MPI and CGA were associated with survival (high risk onco-MPI: HR 5.5, 95%CI 1.25–24.7 *p* = 0.02; fitness at CGA HR 0.552 95% 0.314–0.973; *p* = 0.040) as well as chemotherapy use (HR 0.24, 95% CI 0.11–0.51, *p* < 0.005). **Conclusions:** Both CGA and onco-MPI retain prognostic value for survival in pts with metastatic STS. Pts frail/vulnerable at CGA and pts within the onco-MPI high risk category should be offered an oncogeriatric management approach in order to optimise treatment-related survival and reduce toxicity.

## 1. Introduction

Soft tissue sarcomas (STS) are rare neoplasms accounting for approximately 1% of cancer diagnoses in the adult population [1]. The incidence steadily increases with age, showing a first peak in the adolescent and young adult population and a second peak above 75 years of age [2].

Currently, more than 60% of all cancer diagnoses and 70% of cancer-related deaths occur in elders [3]. With regard to soft tissue and bone sarcomas, older subjects commonly present with a higher prevalence of biologically aggressive subtypes, with higher grade and more advanced disease at diagnosis, and one-year mortality due to sarcoma increases with age [4,5].

Despite the high prevalence in advanced age, older patients are underrepresented in sarcoma clinical trials and evidence-based guidelines are lacking [6].

Older patients with advanced STS, even if highly selected to enter clinical trials, have worse outcomes compared to younger patients; indeed, there is evidence that in routine clinical practice, chemotherapy is denied to a high proportion of patients older than 75 years with advanced STS. Older age (80 y), performance status ≥2 and a high Charlson comorbidity index (≥10) are characteristics associated with the choice of best supportive care [6,7].

Both European and American oncological society guidelines suggest using geriatric tools to evaluate older patients undergoing chemotherapy, and the International Society of Geriatric Oncology (SIOG) has been recommending some form of geriatric assessment since 2005 [8,9,10].

A comprehensive geriatric assessment (CGA) is a multidimensional assessment of an older person which considers health and wellbeing and formulates a plan to address issues which are of concern to the older subject, as well as to their family and caregivers when relevant, and arranges interventions according to the plan. Though there is no standard CGA, most of the used approaches evaluate patients’ abilities in the daily life and instrumental daily life activities, cognitive status, the presence of mood disorders, nutritional status, concomitant medications and comorbidities, the presence and the role of the caregiver and classify patients in either fit or unfit and who might be vulnerable or frail, according to performance in such domains (Figure 1) [11]. CGA adds crucial information on functional assessment, emotional and social aspects of older patients which may compromise quality of life and cancer treatments, and it has long known to be more performant than classical Eastern Cooperative Oncology Group (ECOG) Performance Status (PS) [12,13].

Single items of CGA, such as functional impairment, malnutrition, depressive symptoms, comorbidities, showed to be independently associated to toxicity from chemotherapy and overall survival [14].

Indeed, assessing life expectancy is crucial to help clinicians make fully informed clinical decisions, especially in older patients for whom chronic conditions are competing risk factors for mortality [15].

More recently, an oncological multidimensional prognostic index (onco-MPI) has been developed and validated on the basis of a CGA, which takes into account age, sex, body mass index (BMI), functional impairment, comorbidities, cancer stage, ECOG PS, social status and tumour site. Onco-MPI classifies older patients into three prognostic categories: low, medium and high risk (Figure 2). Onco-MPI has been demonstrated to predict survival probability at one year with a very good discriminatory power and calibration [16].

The predictive and prognostic role of CGA-derived scores has been assessed in several specific cancer types but, as of today, no data on the role of geriatric assessment in decision making for older patients with soft tissue sarcomas are available [17,18,19,20,21].

Our institution has been providing a geriatric assessment to all patients aged 70 years and older, seen as new patients, since 2003. For patients deemed vulnerable or frail at CGA, as well as for fit patients with specific needs, patients are referred to the geriatric service in order to provide geriatric intervention, mainly focused on managing specific impairments, such as polipharmacology, comorbidities and involvement of caregivers for cognitive impairment, as well as for general supportive care.

In light of these considerations, we investigated real-word management and outcomes of older patients with advanced stage soft tissue sarcoma treated at our institution and the prognostic role of CGA and onco-MPI in these patients.

## 2. Patients and Methods

This is a single-centre retrospective study whose aims are describing real-word management and outcomes of older patients with advanced stage soft tissue sarcoma and at assessing the ability of CGA and onco-MPI to predict overall survival and one-year survival in these patients. Consecutive patients with advanced stage STS aged 70 years or older, assessed by means of CGA as per institutional practice and treated at the Istituto Oncologico Veneto from January 2009 to June 2020, were eligible.

The exclusion criteria were: STS pathological subtypes which are not routinely treated with chemotherapy, missing CGA data, single consultation and pts lost to follow-up. Date of diagnosis, age at diagnosis, histological data, PS, site of primary tumour and metastases, first treatment approach, response to treatments, treatment-related toxicities and social, nutritional, psychological and functional aspects were collected from electronic health records and prospectively maintained.

The following CGA domains were considered along with tests used to assess the domain: functional status, through activity of daily living (ADL) and instrumental activity of daily living (IADL), number of comorbid conditions and their severity, through the cumulative illness rating scale (CIRS), living conditions and presence of caregiver, cognitive status through the mini-mental state examination (MMSE) questionnaire, emotional status through the geriatric depression scale (GDS), polipharmacy, nutritional status through the BMI and mini-nutritional assessment (MNA) and the presence of geriatric syndromes [22,23,24,25,26,27,28]. Patients were classified into risk categories according to both Balducci’s criteria and onco-MPI, as previously published (Table 1), with the three categories of fit, vulnerable and frail for Balducci’s criteria and the three categories of low risk (score 0.0–0.46), intermediate risk (score 0.47–0.63) and high risk (score 0.64–1.0) for the onco-MPI (Table 2) [11,16].

Overall survival (OS) was estimated with the Kaplan–Meier method and compared with the log-rank test and Cox proportional hazards method for multivariate analysis. Cox’s proportional hazard assumptions have been graphically verified and are respected for the variables considered in the model. OS was calculated from diagnosis of metastatic disease to death for any cause. The survival status of patients lost to follow-up was obtained through demographic registries.

The study was approved by the Ethics Committee of the Istituto Oncologico Veneto. Statistical analysis was performed with R software version 3.4.3.

## 3. Results

A total of 168 patients with a diagnosis of advanced/metastatic or locally advanced STS, aged 70 years or older, were identified, of whom 30 were excluded due to disease characteristics. Other patients (N = 37) were excluded because of missing data for CGA and/or onco-MPI, or because of a single access to the institution for consultation. In total, one hundred and one patients were eligible (Figure 1).

Patients’ characteristics are shown in Table 3.

The median age was 77 years (range 70–91 years), with 66 patients (65.3%) being aged 75 years and older. Primary tumour sites were the extremities or trunk for 44 patients (43.5%); the most frequent histological subtypes were liposarcoma (27 patients) and leiomyosarcoma (26 patients) and the most frequent metastatic site was the lungs (64 patients, 63.3%).

Out of 101 patients, 76 received chemotherapy (75.3%), which was anthracycline-based for 46 patients.

According to the CGA categories, 39 patients were fit (38.6%), 46 were vulnerable (45.6%), 16 were frail (15.8%); according to onco-MPI, 87 patients (86.1%) were in the high-risk category, 14 (13.9%) were in the intermediate risk and no patients were in the low-risk category.

Chemotherapy was administered to 82% of the patients in the fit group, in 80% of patients in the vulnerable group and 43% of frail patients (*p* = 0.016). Anthracyclines were used in 23 (58.9%), 21 (45.1%) and 2 (12.5%) patients of the fit, vulnerable and frail group, respectively (*p* = 0.07) (Figure 2). Toxicities and upfront dose reductions due to toxicities are reported in Table 4.

First line chemotherapy was administered to 85.7% of patients in the intermediate risk group, according to onco-MPI, and in 74.1% of high-risk group; an anthracycline- based chemotherapy was administered in 71.4% of patients in the intermediate risk group and in 42.3% in the high-risk group (*p* = 0.046) (Figure 2).

Toxicities and dose reduction are also reported in Table 4.

The median follow-up was 32.4 months (95% CI 0–202); median OS was 13.8 months (95% CI 11.3–17.7).

According to CGA categories, the median OS of fit patients was 19.53 months (95% CI 15.23–36.8), compared to 12.83 months (95% CI 9.7–17.5) in vulnerable and 7.75 months (95% CI 2.73–30) in frail patients (*p* = 0.005) (Figure 3).

In univariate analysis, patients with ECOG PS ≥ 2 showed the worst survival, HR 2.34 *p* < 0.001; fit patients (HR 0.47, 95% CI 0.29–0.75) and patients who received first line chemotherapy (HR 0.25, 95% CI 0.15–0.4) had better survival (*p* < 0.005) (Table 5). Receiving anthracycline-based chemotherapy was not associated with an advantage in survival (HR = 0.84, CI 0.5459–1.294, *p* = 0.429).

The onco-MPI score was a predictor of one-year survival; in fact, patients with intermediate risk onco-MPI did not reach a median one-year OS, while patients in the high-risk group had a median OS of 11.5 months (95% CI 9.7–NA); log rank test 5.7, *p* = 0.02. Figure 4.

In the multivariate analysis, fitness at CGA and chemotherapy receipt were associated with better survival (HR 0.55, 95% CI 0.3–0.97, *p* = 0.04; and HR 0.36, 95% CI 0.19–0.67, *p* = 0.001, respectively), as shown in Table 6.

Despite its good predictiveness of one-year survival, onco-MPI was not associated with global OS in the multivariate analysis.

When analysing the predictivity of one-year survival, onco-MPI and chemotherapy receipt were correlated with survival in the univariate analysis. The multivariate analysis confirmed the correlation between the onco-MPI high-risk group and worse survival (HR 5.5, 95%CI 1.25–24.7 *p* = 0.02) as well as the correlation of chemotherapy receipt and better survival (HR 0.24, 95% CI 0.11–0.51, *p* < 0.005), as shown in Table 6.

## 4. Discussion

This study provides data on the treatment and outcomes in an unselected real-world population of older patients with advanced/metastatic STS. To the best of our knowledge, the present study is the first to investigate the role of CGA and of the CGA-derived onco-MPI as prognosticators in older patients with metastatic STS. In the setting of advanced stage disease, chemotherapy has a palliative intent. In fact, chemotherapy has been shown to improve survival in patients with STS, though older patients experience a higher rate of toxicity [29,30] and benefits need to be thoroughly weighed against the risks in the frame of competing risks for mortality. Indeed, life expectancy estimation in older patients is crucial and tools to improve prognosis assessment in older patients with cancer are of utmost importance.

Age has been shown to be among the predictors of toxicity from doxorubicin in a retrospective analysis of the European Organization for Research and Treatment of Cancer (EORTC) Soft Tissue and Bone Sarcoma Group database [31].

Moreover, for older patients, quality of life might be more worthwhile than the prolongation of life. Indeed, a recent Dutch study showed that patients with advanced stage sarcoma, aged less than 40 years, prioritised length of life, whereas two-thirds of patients aged ≥65 years felt that quality of life was equally or more important than length of life [32].

In our cohort of unselected patients with metastatic STS in the real-world practice, with median age being 77 years, global median OS was 13.8 months.

Our study showed CGA to be predictive of overall survival, and confirmed Onco-MPI to predict one-year survival.

Palliative chemotherapy may have a role for some older patients, as an American and French study also showed, with the mOS of patients managed with systemic therapy being 10.9 months versus 5.3 months for patients managed with best supportive care [7]. Our data show that fit patients may reach their median OS as high as 19.5 months. These figures are comparable to those derived from randomised trials of chemotherapy in patients with sarcoma, in which the median age is lower, such as in the Announce trial, in which the median age was 56.9 years [33]. Even in randomised clinical trials, specifically designed for older patients with sarcoma, with the age cut-off set nonetheless at 65 years, mOS ranged from 16.7 months (evofosfamide vs. doxorubicin trial) to 12.3 months (trofosfamide vs. doxorubicin trial) to 14.3 months (pazopanib vs. doxorubicin trial) [34,35,36].

Frail patients had a significantly shorter mOS of 7.75 months, as well as vulnerable patients, whose survival was 12.83 months. These data in frail older patients compare favourably with the literature data from retrospective studies, such as the English experience, in which the mOS is 6.5 months [37].

Our results therefore confirm the strong prognostic value of CGA for mOS in older patients with advanced STS, as already demonstrated for patients treated with chemotherapy for many other solid tumour types.

Unlike results from a previous study, in which age and PS were independently associated with survival, in our cohort of patients, these two variables, not other tumour single characteristics, were independent prognostic factors, confirming that a full assessment is more predictive than single variables, in line with studies in other cancer types [7,21,38,39,40]

Better survival rates observed in fit patients may have multiple explanations: fit patients are expected to be in better general conditions with less comorbidities, and this more likely influences a clinician’s decision to propose first line chemotherapy. The receipt of chemotherapy is the only variable that retains significance in the multivariate analysis in our study. Fit patients were treated with chemotherapy in a significantly higher proportion compared to frail patients (82% vs. 43%), and more often the regimen included anthracyclines (58.9% vs. 12.5%). The proportion of patients receiving chemotherapy is not significantly different when comparing fit and vulnerable patients (82% vs. 78.2%), yet the use of anthracycline-based regimens is significantly higher in fit patients (58.9% vs. 45.6%). Interestingly, our data showed that anthracycline-based chemotherapy did not provide a clear-cut benefit in older patients, and this is the rationale for ongoing randomised trials assessing first line use of doxorubicin vs. metronomic cyclophosphamide in older patients with metastatic STS (METROPHOLYS trial NCT04656262, TOLERANCE trial NCT04780464).

Our data also showed that vulnerable patients did not experience a higher rate of toxicity and dose reduction compared to fit patients, despite the evidence from some trials in which chemotherapy toxicity rates increased with geriatric impairments [38,41]. Such findings might be due to a higher proportion of patients in the fit group receiving anthracyclines in relation to comorbidity. Moreover, geriatric interventions offered to patients with impaired items at CGA might have had an impact in reducing the toxicity among vulnerable and frail patients.

As stated, vulnerable or frail patients at CGA were more often offered geriatric intervention to manage specific impairments, such as polipharmacology, comorbidities and cognitive impairment, as well as more general supportive care.

Indeed, geriatric interventions have been shown to reduce chemo-related toxicities in two recently published trials, randomising older patients to either receive chemotherapy as standard practice or to receive chemotherapy and CGA-driven geriatric intervention (GAIN study; GAP70+ study) [42,43] and to improve quality of life (INTEGERATE study) [44].

Onco-MPI intermediate and high-risk categories are the most represented in our study cohort, and this is likely due to metastatic stage conferring a higher risk in the stage onco-MPI domain. Moreover, patients with sarcoma were a small number in the development cohort of the onco-MPI, thus included under the “other” category. Nonetheless, onco-MPI in this study confirmed its role as a one-year survival predictor in patients with advanced stage STS, with patients in the intermediate risk category who did not reach the median survival and those in the high-risk category having a median survival of 11.5 months (*p* = 0.02).

The onco-MPI score in this population allowed a better prediction of mortality at one year, while the CGA impact is higher in distinguishing the unfit group of patients as vulnerable and frail. Therefore, the use of both CGA and onco-MPI could better stratify older patients as candidates for first line chemotherapy.

This study has some limitations residing mainly in its retrospective design, long accrual period, low number of patients due to the rarity of disease and the single centre experience. The low number of patients and the large number of variables could also have caused a moderate overfitting of the model, despite the number of events being consistent. Large multicentric and prospective trials, either randomised or observational, might therefore provide more solid and robust data.

As a matter of fact, decision making for vulnerable and frail patients remains an unmet need. Further prospective studies within this target population are needed, evaluating the role of geriatric intervention on chemo-related toxicities, quality of life and survival, though setting up and conducting such trials is challenging due to the rarity of the disease and the not-so-widespread *geriatrisation* of medical oncology units.

## 5. Conclusions

Benefit/risk balance in the approach to older patients with advanced or metastatic STS must be accurately considered. In our study, geriatric assessment has been confirmed to be prognostic and predictive also in rare diseases, such as sarcoma, and can better inform clinicians compared to simply considering chronological age or PS. Prospective studies incorporating geriatric parameters in the decision making for older STS are ongoing.

## Figures and Tables

**Figure 1 cancers-15-01043-f001:**
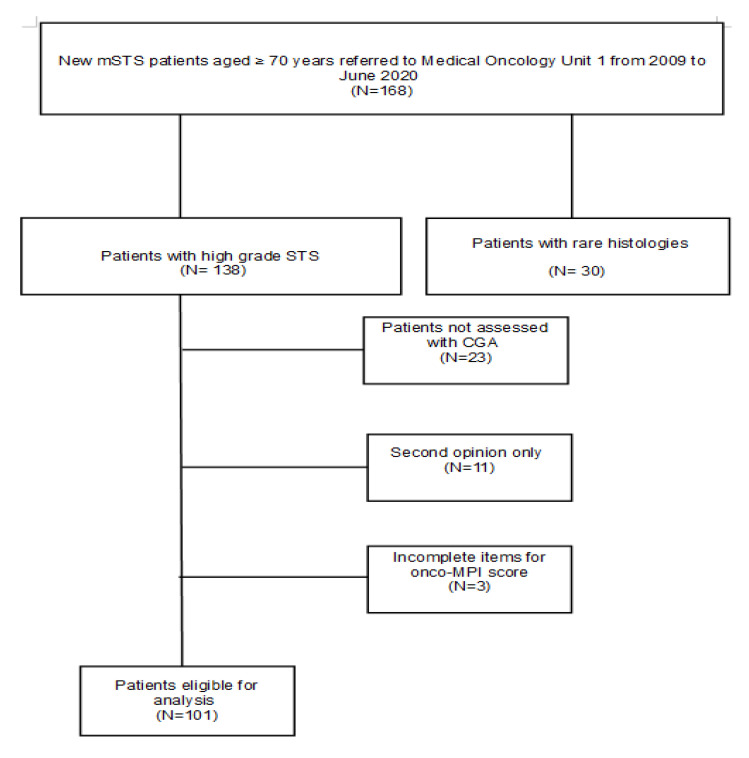
CONSORT flow diagram.

**Figure 2 cancers-15-01043-f002:**
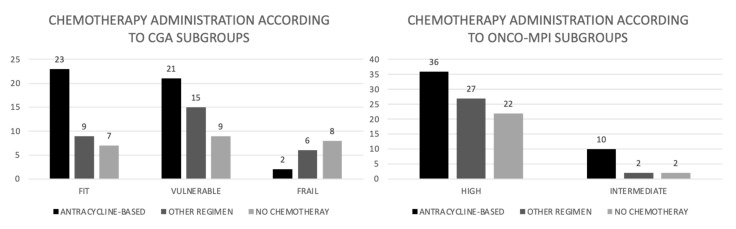
Chemotherapy administration according to CGA and onco-MPI subgroups (absolute numbers).

**Figure 3 cancers-15-01043-f003:**
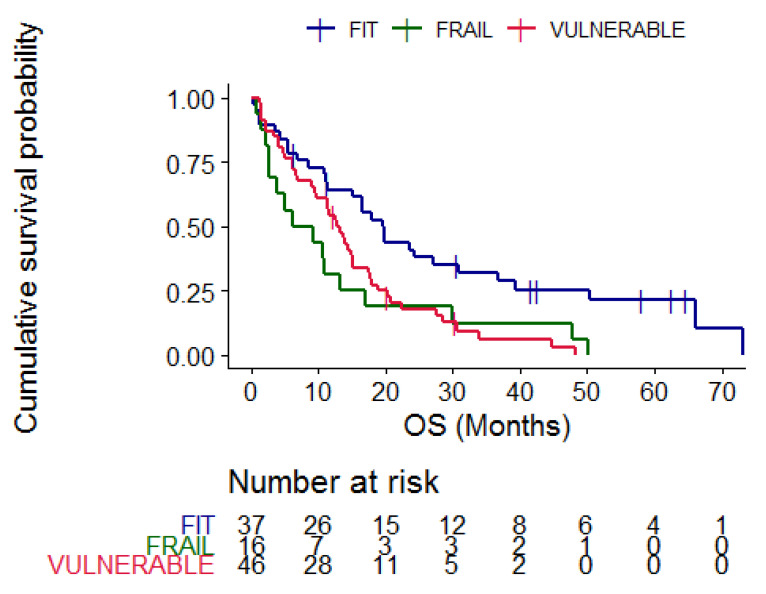
Kaplan–Meier curves representing overall survival according to CGA categories, blue = fit patients, red = vulnerable patients and green = frail patients. *p* = 0.005.

**Figure 4 cancers-15-01043-f004:**
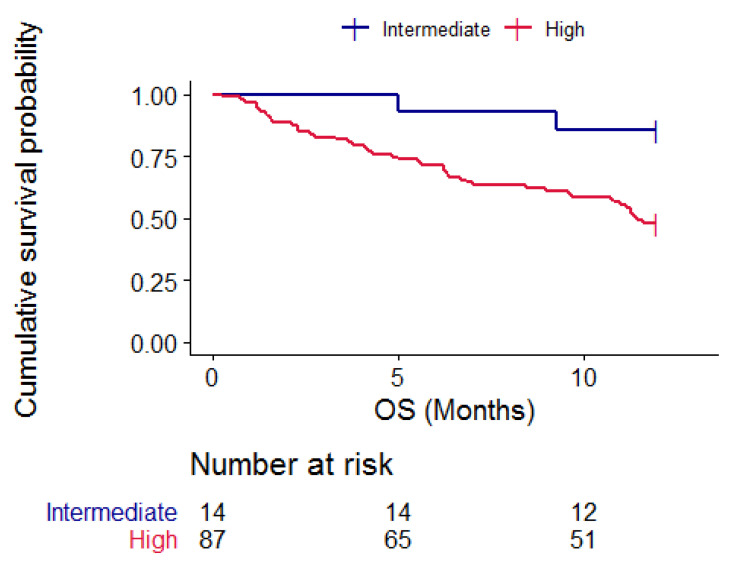
Kaplan–Meier curves representing one-year OS according to onco-MPI categories, blue = intermediate risk patients and red = high-risk patients. *p* = 0.02.

**Table 1 cancers-15-01043-t001:** Classification of patients according to Balducci’s criteria.

Fit	Vulnerable	Frail
- No functional dependence in ADLs and IADLs - No relevant comorbidities - No geriatric syndromes	- Dependence in one or more IADLs but not ADLs - Comorbidities present but manageable and not life-threatening - Mild memory disorder and/or depression - No geriatric syndromes	- Age ≥ 85 years - Dependence in one or more items of ADLs - Geriatric syndromes - Three or more grade 3 comorbidities or one grade 4 comorbidity

Legend: ADL = Activities of Daily Living; IADL = Instrumental Activities of Daily Living.

**Table 2 cancers-15-01043-t002:** Onco-MPI algorithm [16].

Domains Onco-MPI	Category	Coefficient
Age (≥70 years)	continuous variable	0.0473
Sex	0 F	0
	1 M	0.01706
BMI (Kg/m^2^)	continuous variable	−0.09782
ADL	continuous variable	−0.07717
IADL	continuous variable	0.04983
Performance status	continuous variable	0.70607
N° of severe comorbidities (CIRS)	continuous variable	−0.1296
Cancer stage	1	0
	2	1.11712
	3	0.74957
	4	1.80828
Tumour site	other	0
	Breast	−1.93081
	Colorectal	−1.03025
	Lung	0.36265
	Prostate	−1.57998
	other GU	0.19956
MMSE	<24	0
	≥24	0.0627
N° of drugs	continuous variable	−0.01218
Caregiver	No	0
	Yes	0.21035

Legend: BMI = Body Mass Index; ADL = Activities of daily living; IADL = Instrumental activities of daily living; CIRS = Cumulative illness rating scale; GU = genitourinary; MMSE = Mini-mental state examination.

**Table 3 cancers-15-01043-t003:** Patient’s characteristics (N = 101).

Characteristics	Categories	N (%)
Age		77 (±4.3)
Sex	Male/female	51/42
Histology	Liposarcoma	26 (27.9%)
	Leiomyosarcoma	24 (25.8%)
	UPS	14 (15.0%)
	Other	29 (33.1%)
Primary site	Extremities/trunk	38 (40.8%)
	Abdomen	29 (31.1%)
	Chest	6 (0.6%)
	Other	20 (21.5%)
Metastatic sites	Lung	24 (25.8%)
	Other	35 (37.6%)
	Lung and other	34 (36.6%)
PS ECOG	0–1	76 (75.2%)
	≥2	25 (24.8%)
CIRS	0–2	
	>2	
N. of medications	≤3	52 (51.4%)
	>3	49 (48.6%)
1st line chemotherapy	Anthracycline-based	44 (47.3%)
	CM	11 (11.8%)
	Other	15 (16.1%)
CGA	Fit	39 (38.6%)
	Vulnerable	46 (45.6%)
	Frail	16 (15.8%)
Onco-MPI	Low	0 (0%)
	Intermediate	14 (13.9%)
	High	87 (86.1%)

Legend: ADL = Activities of daily living; CGA = Comprehensive geriatric assessment; CIRS = Cumulative illness rating scale; ECOG = Eastern cooperative oncology group; PS = Performance status.

**Table 4 cancers-15-01043-t004:** Rates of severe toxicities, upfront dose reduction and access to second line treatment according to CGA category and onco-MPI risk groups (% of pts).

N. Patients Receiving First Line CT	FIT (32 pts)	Vulnerable (36 pts)	Frail (8 pts)	Intermediate Risk (12 pts)	High Risk (63 pts)
G3-G4 Toxicity	40.6%	33.3%	51.1%	58.3%	36.5%
Upfront Dose Reduction	34.4%	44.4%	42.8%	66%	39.4%
Dose Reduction due to Toxicity	12.5%	8.3%	12.5%	33.3%	15.8%
Second Line CT	38.4%	34.7%	12.5%	66%	58.7%

Legend: CGA = Comprehensive geriatric assessment; CT = Chemotherapy; G3–G4 = Grade 3 or 4 according to Common Terminology Criteria for Adverse Events.

**Table 5 cancers-15-01043-t005:** Univariate and multivariate analysis for overall survival.

	**Univariate Analysis for Overall Survival**				
Variable		HR	Confidence Interval		*p* Value
Sex	M	0.378	0.948	12.250	0.0858
Histology	Leiomyosarcoma	0.681	0.418	1.111	0.124
Age	75–80	1.163	0.702	1.927	0.557
Age	≥80	1.025	0.605	1.736	0.928
Location metastasis	Lung	0.822	0.502	1.347	0.437
Location of primary tumour	Extremities/trunk	0.940	0.581	1.521	0.800
Location of primary tumour	Other	0.327	0.549	1.761	0.955
PS at metastatic disease	≥2	2.345	1.449	3.794	<0.001
Comorbidity grade according to CIRS	3–4	1.511	0.898	2.543	0.120
CGA	Fit	0.471	0.293	0.755	0.001
Onco-MPI	High-risk	1.300	0.670	2.526	0.438
First line chemotherapy	Yes	0.251	0.153	0.414	<0.001
	**Multivariate Analysis for Overall Survival**				
Variable		HR	Confidence Interval		*p*-Value
Sex	Male	1.4047	0.826	2.388	0.2096
Histology	Leiomyosarcoma	0.7135	0.396	1.285	0.2604
Age	≥80	0.718	0.396	1.301	0.2746
Age	75–80	0.7542	0.406	1.399	0.3707
Location of metastasis	Lung	0.774	0.441	1.357	0.3713
Location of primary tumour	Retroperitoneum	1.0266	0.565	1.866	0.9313
Location of primary tumour	other	0.9905	0.516	1.900	0.9771
PS at metastatic disease	≥2	1.5004	0.760	2.962	0.2424
Comorbidity grade according to CIRS	3–4	1.2159	0.664	2.226	0.5263
First line chemotherapy	Yes	0.3634	0.195	0.676	0.0014
CGA	Fit	0.5527	0.314	0.973	0.040

Legend: CGA = Comprehensive geriatric assessment; CIRS = Cumulative illness rating scale; ECOG = Eastern cooperative oncology group; PS = Performance status.

**Table 6 cancers-15-01043-t006:** Univariate analysis and multivariate analysis for one-year survival.

	**Univariate Analysis**				
Variable		HR	Confidence Interval		*p* Value
Sex	M	1.481	0.822	2.667	0.191
Histology	Leiomyosarcoma	0.633	0.306	1.309	0.218
Age	75–80	0.641	0.322	1.280	0.208
Age	≥80	0.710	0.356	1.416	0.331
Location metastasis	Lung	0.761	0.387	1.496	0.428
Location of primary tumour	Retroperitoneum	1.687	0.852	3.341	0.134
Location of primary tumour	Other	2.086	1.017	4.275	0.045
PS at metastatic disease	≥2	2.754	1.526	4.971	<0.001
Comorbidity grade according to CIRS	3–4	1.043	0.502	2.171	0.91
Onco-MPI	High-risk	4.743	1.150	19.56	0.03
First line chemotherapy	Yes	0.193	0.106	0.354	<0.001
	**Multivariate Analysis**				
Variable		HR	Confidence Interval		*p* Value
Sex	Male	1.4579	0.666	3.188	0.3450
Histology	Leiomyosarcoma	0.5210	0.211	1.287	0.1574
Age	≥80	0.3968	0.174	0.906	0.0283
Age	75–80	0.5406	0.249	1.175	0.1203
Location of metastasis	Lung	0.5860	0.268	1.280	0.1802
Location of primary tumour	Extremities/trunk	2.0642	0.952	4.478	0.0666
Location of primary tumour	Other	2.2186	0.959	5.129	0.0624
PS at metastatic disease	≥2	2.0196	0.889	4.586	0.093
Comorbidity grade according to CIRS	3–4	0.8535	0.366	1.989	0.7136
First line chemotherapy	YES	0.2405	0.113	0.512	0.0002
Onco-MPI	High-risk	5.5682	1.251	24.793	0.0242

Legend: CGA = Comprehensive geriatric assessment; CIRS = Cumulative illness rating scale ECOG = Eastern cooperative oncology group; PS = Performance status.

## Data Availability

Data presented in this study are available on request from the corresponding author.

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
