# Peer review of "Management and Outcomes of Older Patients (Age ≥ 70 Years) with Advanced Soft Tissue Sarcoma and Role of Geriatric Assessment and Oncological Multidimensional Prognostic Index (Onco-MPI) in a Real-World Setting"

_cancers, 2023, doi:10.3390/cancers15041043_

Round 1
Reviewer 1 Report
Thank you for the opportunity to review "Role of geriatric assessment and oncological multidimensional prognostic index (onco-MPI) in older patients (age ≥70 years) with advanced soft tissue sarcoma in a real-world setting".
The topic is relevant and the research group has access to relevant and thoroughly collected data about patients with soft tissue sarcoma. The data has been collected through 9,5 years in a single center in Italy and the group has gathered information about 101 patients – app. 10 patients per year.
My overall impression is that the draft still needs to be worked through again – both to be more precise in aim/structure and to improve the overall language with e.g. missed punctuations, commas, very long sentences which makes the text difficult to understand/read (e.g, lines 81-84).
Title: Difficult to understand what is meant by "role" – is it the ability to find frail patients (sceening-tool)? Or is it to describe distribution of frailty in the population? Or is it to find association between level of frailty and course of disease (toxicity and death)? Or a combination? Please be more precise.
Frailty/CGA
The definition of frailty is deficient. From my point of view (as geriatrician) the use of the term Comprehensive Geriatric Assessment is misunderstood in the abstract and also as used term it the rest of the text. E.g. in abstract line 45-46: the point of a CGA is not to classify patients into 3 categories – it is to make a list of issues of importance for the patient's health and what to do about it. In the everyday clinical practice geriatric patients are not classified – though this often is the case and needed when doing research.
CGA is correctly explained in lines 89-92. But what is performed in this study is not a CGA. It is a Geriatric Assessment (GA) – a term (correctly) used in the title. To become comprehensive, it must be performed by a geriatrician or geriatric team and actions must be executed. In this paper, you get the impression that some test has been performed by someone (not stated).
Onco-MPI is an interesting measure and I would suggest it to be the main topic investigated. The Balducci Criteria are >20 years old and does not seem to be un use or relevant anymore as a measure. SIOG recommends use of a number of geriatric screening tools and e.g G8, mG8 or VES13 would be more common and precise than the assessments done in the Balducci-classification (e.g. not everybody can make the assessment of whether a comorbidity is "relevant" or "manageable"). It does not seem as if the topic is to evaluate how the two scores correspond to each other – but rather how each of them can predict the course of the disease?
It is important to highlight that a screening-tool-result can be a prognostic for a given condition – but that the screening is not validated to or meant to "rule patients out" – it should always be followed up by a comprehensive evaluation if a patient is screened to be "frail" to ensure a holistic and appropriate treatment offered.
Abstract:
There is no aim or purpose mentioned in the abstract. Based on the results and the conclusion one can speculate about what is was – but should be stated clearly.
Background:
Though the background includes important points about frail older patients with cancer it does not lead up to a specific aim. Also, the references are not updated (see below). Needs more structure and update.
Aim: as in title – need to specify the aim(s) to the reader. The current text does not help the reader to get a clear understanding of a systematic evaluation of "role" or to easily read and understand the following text and especially not the very long result part.
Method:
In Lines 128-132 efforts are made to explain the CGA-domains (on which onco-MPI is build) and that are used in the Balducci-score. There is a differences between a domain and a test which is not clear in the paper. CGA domains can be eg:
Domain: Multimorbidity and the corresponding "test" can be CIRS-G or CCI.
Domain: Polypharmacy with corresponding "test": number of prescription drugs.
Domain: Nutrition – with corresponding "test" MNA, BMI
Domain: Kognitive/mental: corresponding tests – MMSE and GDS.
This study most likely has performed tests corresponding to key-domains in CGA – but the method part lacks a clear description of what was done and by whom. A regular table or just explanation of domains and corresponding tests would be very helpful.
Figures and tables:
Figures and tables are plenty and need more work on both appearance and legends. E.g. figure 1 is presented almost like a very early draft with draft-signs and also wrong(?) year – states 2011 whereas the text state the study period to be from 2009.
No figure/table legends are self-explaining. It would be helpful if the legend tells what the table is about. Especially with so many figures/tables. E.g. table 3: patient's characteristics in a cohort of 101 pt.s with STS (or even more specifically). All abbreviations should be explained in each of the tables. Also in table 3, the lines do not correspond with titles so it difficult to see what numbers belong to what characteristic and category. In table 3 the number of patients reported does not correspond with the text. In the text the population consists of 101 patients. In the table "Sex" adds up to 93 persons and the same does "Histologies". In "Primary site" the number is 152 patients and "metastatic sites" 93 – only few characteristics adds up to 101. It would be helpful for the reader if e.g. in chemotherapy it is stated how many pts did not receive chemo – then you can see that the correct number is included.
There are many tables and figures. Are they all necessary or should they be written in text? Now the text repeats all the numbers. E.g. in lines 161 onwards the text is "reading the table aloud" and does not add anything.
Discussion:
Does not read easy. Should be structured and aligned with the method and result section (that also needs structure).
Language: not clear –see example in lines 259-263 -. please use commas/punctuations and make clear.
Conclusion:
Please revise in the language in the whole article to be clearer. Line 303: ain?
Since the aim is difficult to understand in the beginning it is also not clear if there is an answer to the aim(s). The first sentence in the conclusion is not a conclusion from the study results but rather a background note.
The conclusion could be restricted to only line 306 from "this study confirms the importance of inclusion of geriatric assessment/screening when considering… "
References:
Overall, the references are rather old when regarding the current rapid increase in onco-geriatric literature and studies. Of course basic studies like the Balducci study from 2000 is relevant. But it would be relevant to update the reference list/text to include new large studies of sceening-tools (if screening is an aim) (e.g. PROGNOSIS-G8 https://pubmed.ncbi.nlm.nih.gov/34176752/ - just an example) and studies of toxicity and adherence to care-plan in older cancer patients according to CGA (e.g. the GAIN study https://pubmed.ncbi.nlm.nih.gov/34591080/ , the GAP70+ study https://pubmed.ncbi.nlm.nih.gov/34741815/ or GERICO https://pubmed.ncbi.nlm.nih.gov/33828260/ as examples).
In some references the study, population are dating back to year 1991. One would think that significante improvements have been done in the field of chemotherapy since and that it is not a comparable population. But I am not oncologist and do not know of this topic specifically.
Author Contribution:
It is notable that the study is performed of 15 authors (14 oncologists and 1 geriatrician). It is not reported how the authors have contributed substantially to the work and if everybody on this extensive list are involved to an extend that justifies authorship. Must be updated.
I hope that the authors will revise the article so the results can be published.
Author Response
Dear Editor
Thank you for the opportunity to revise and resubmit our manuscript entitled “Role of geriatric assessment and oncological multidimensional prognostic index (onco-MPI) in older patients (age ≽70 years) with advanced soft tissue sarcoma in a real-world setting” to your journal.
We would like to thank the reviewers for their time and for providing us with helpful and constructive comments.
Here below, we have answered the referees’ questions point-by-point (answers in bold italics).
REVIEWER #1
Thank you for the opportunity to review "Role of geriatric assessment and oncological multidimensional prognostic index (onco-MPI) in older patients (age ≥70 years) with advanced soft tissue sarcoma in a real-world setting". The topic is relevant and the research group has access to relevant and thoroughly collected data about patients with soft tissue sarcoma. The data has been collected through 9,5 years in a single center in Italy and the group has gathered information about 101 patients – app. 10 patients per year.
- My overall impression is that the draft still needs to be worked through again – both to be more precise in aim/structure and to improve the overall language with e.g. missed punctuations, commas, very long sentences which makes the text difficult to understand/read (e.g, lines 81-84).
As advised, we have performed a language proofing and corrected grammar / language errors throughout the paper.
- Title: Difficult to understand what is meant by "role" – is it the ability to find frail patients (screening-tool)? Or is it to describe distribution of frailty in the population? Or is it to find association between level of frailty and course of disease (toxicity and death)? Or a combination? Please be more precise.
Indeed, in the oncology practice there is no routine use of geriatric assessment tools, which might inform clinical decisions by allowing identification of vulnerabilities which in turn might be corrected and therefore might serve to improve outcomes. In this sense we used the word “role”, since we are evaluating different tools to assess older patients, tools that might have different impact and effectiveness in optimization of oncological treatment.
3.Frailty/CGA
The definition of frailty is deficient. From my point of view (as geriatrician) the use of the term Comprehensive Geriatric Assessment is misunderstood in the abstract and also as used term it the rest of the text. E.g. in abstract line 45-46: the point of a CGA is not to classify patients into 3 categories – it is to make a list of issues of importance for the patient's health and what to do about it. In the everyday clinical practice geriatric patients are not classified – though this often is the case and needed when doing research. CGA is correctly explained in lines 89-92. But what is performed in this study is not a CGA. It is a Geriatric Assessment (GA) – a term (correctly) used in the title. To become comprehensive, it must be performed by a geriatrician or geriatric team and actions must be executed. In this paper, you get the impression that some test has been performed by someone (not stated).
In the manuscript we use the term “Comprehensive Geriatric Assessment” in light of its very first definition in Oncology proposed by Lodovico Balducci in the early 2000 and that was the basis the very few oncologists who did perform geriatric assessment worked on (Balducci L. Management of Cancer in the Older Person: A Practical Approach. Vol 5.; 2000. doi:10.1634/theoncologist.5-3-224), with the gross classification in frail / vulnerable / fit subjects in relation to suitability for anticancer directed treatment. We agree with the reviewer that, indeed, the one performed in the clinical routine in oncology is not quite “Comprehensive” though we are moving forward. Indeed, Geriatric Assessment in our Institution is performed by psychologists trained in geriatric assessment and patients with frailty, and patients identified as frail are now invited to be assessed by geriatricians for eventual recommendations.
4.Onco-MPI is an interesting measure and I would suggest it to be the main topic investigated. The Balducci Criteria are >20 years old and does not seem to be un use or relevant anymore as a measure. SIOG recommends use of a number of geriatric screening tools and e.g G8, mG8 or VES13 would be more common and precise than the assessments done in the Balducci-classification (e.g. not everybody can make the assessment of whether a comorbidity is "relevant" or "manageable"). It does not seem as if the topic is to evaluate how the two scores correspond to each other – but rather how each of them can predict the course of the disease?”. It is important to highlight that a screening-tool-result can be a prognostic for a given condition – but that the screening is not validated to or meant to "rule patients out" – it should always be followed up by a comprehensive evaluation if a patient is screened to be "frail" to ensure a holistic and appropriate treatment offered.
We agree with the reviewer that Balducci’s criteria are quite old and new tools are increasingly been used in oncology. Yet, they have long been applied in oncology, and we wanted to look for their capacity to predict disease course and/or toxicity in the case of rare neoplasms such as sarcomas, and whether OncoMPI in this particular setting was an index which kept its reliability. Indeed, the OncoMPI was validated to predict 1 year mortality risk in several cancers, mainly carcinomas, with sarcomas being very few in both the development and the validation cohort. As pointed out, we were not interested in comparing the two tools but in defining the prognostic value of each of them.
More recently, we started screening patients by means of the G8 questionnaire, which – being purely a rapid screening test – has a high sensitivity but low specificity for identifying frail patients. Thus, only patients with impaired G8 score are now submitted for a CGA/OncoMPI scoring. Our study, being retrospective, does not include this innovation, which will be definitely taken into account in a prospective study which we plan to start
- Abstract: There is no aim or purpose mentioned in the abstract. Based on the results and the conclusion one can speculate about what is was – but should be stated clearly.
Thank you for the comment, the abstract has been updated accordingly.
- Background: Though the background includes important points about frail older patients with cancer it does not lead up to a specific aim. Also, the references are not updated (see below). Needs more structure and update. Thank you for the comment, the background has been amended taking into account these suggestions.
- Aim: as in title – need to specify the aim(s) to the reader. The current text does not help the reader to get a clear understanding of a systematic evaluation of "role" or to easily read and understand the following text and especially not the very long result part.
The aim of our study was to describe real-word management and outcomes of older patients with advanced stage soft tissue sarcoma (since no benchmark data are available for this population), along with assessing the ability of CGA and oncoMPI to predict survival in these patients. Text has been amended accordingly.
- Method:
In Lines 128-132 efforts are made to explain the CGA-domains (on which onco-MPI is build) and that are used in the Balducci-score. There is a differences between a domain and a test which is not clear in the paper. CGA domains can be eg:
Domain: Multimorbidity and the corresponding "test" can be CIRS-G or CCI.
Domain: Polypharmacy with corresponding "test": number of prescription drugs.
Domain: Nutrition – with corresponding "test" MNA, BMI
Domain: Kognitive/mental: corresponding tests – MMSE and GDS.
Thank you for the comment, the manuscript has been amended accordingly.
This study most likely has performed tests corresponding to key-domains in CGA – but the method part lacks a clear description of what was done and by whom. A regular table or just explanation of domains and corresponding tests would be very helpful.
Thank you for the comment, the manuscript has been updated.
- Figures and tables:
Figures and tables are plenty and need more work on both appearance and legends. E.g. figure 1 is presented almost like a very early draft with draft-signs and also wrong(?) year – states 2011 whereas the text state the study period to be from 2009.
Thank you for the comment, the manuscript has been updated.
No figure/table legends are self-explaining. It would be helpful if the legend tells what the table is about. Especially with so many figures/tables. E.g. table 3: patient's characteristics in a cohort of 101 pts with STS (or even more specifically). All abbreviations should be explained in each of the tables. Also in table 3, the lines do not correspond with titles so it difficult to see what numbers belong to what characteristic and category. In table 3 the number of patients reported does not correspond with the text. In the text the population consists of 101 patients. In the table "Sex" adds up to 93 persons and the same does "Histologies". In "Primary site" the number is 152 patients and "metastatic sites" 93 – only few characteristics adds up to 101. It would be helpful for the reader if e.g. in chemotherapy it is stated how many pts did not receive chemo – then you can see that the correct number is included.
There are many tables and figures. Are they all necessary or should they be written in text? Now the text repeats all the numbers. E.g. in lines 161 onwards the text is "reading the table aloud" and does not add anything.
Thank you for the comment, the manuscript has been updated, with legends and abbreviations explained and we combined some tables reducing total number.
- Discussion:
Does not read easy. Should be structured and aligned with the method and result section (that also needs structure).
Discussion has been revised, with language editing. We hope it is more consistent now and easier to read.
Language: not clear –see example in lines 259-263 -. please use commas/punctuations and make clear.
Discussion has been revised, with language editing.
- Conclusion:
Please revise in the language in the whole article to be clearer. Line 303: ain?
Text has been revised, with language editing.
Since the aim is difficult to understand in the beginning it is also not clear if there is an answer to the aim(s). The first sentence in the conclusion is not a conclusion from the study results but rather a background note.
The conclusion could be restricted to only line 306 from "this study confirms the importance of inclusion of geriatric assessment/screening when considering… "
Thank you for the comment, the manuscript has been updated
- References:
Overall, the references are rather old when regarding the current rapid increase in onco-geriatric literature and studies. Of course basic studies like the Balducci study from 2000 is relevant. But it would be relevant to update the reference list/text to include new large studies of sceening-tools (if screening is an aim) (e.g. PROGNOSIS-G8 https://pubmed.ncbi.nlm.nih.gov/34176752/ - just an example) and studies of toxicity and adherence to care-plan in older cancer patients according to CGA (e.g. the GAIN study https://pubmed.ncbi.nlm.nih.gov/34591080/ , the GAP70+ study https://pubmed.ncbi.nlm.nih.gov/34741815/ or GERICO https://pubmed.ncbi.nlm.nih.gov/33828260/ as examples).
In some references the study, population are dating back to year 1991. One would think that significant improvements have been done in the field of chemotherapy since and that it is not a comparable population. But I am not oncologist and do not know of this topic specifically.
Thank you, we revised the reference section and updated some bibliography items. Yet, it must be said that still to date advanced-stage sarcoma treatment is based on doxorubicin which is a 50-year old drug and no major improvements have been made since, despite several attempts in finding better first line regimens.
- Author Contribution:
It is notable that the study is performed of 15 authors (14 oncologists and 1 geriatrician). It is not reported how the authors have contributed substantially to the work and if everybody on this extensive list are involved to an extend that justifies authorship. Must be updated.
All the authors and co-authors of this manuscript (except for the statistician) are part of the multidisciplinary team involved in sarcoma patients clinic (oncologists, surgical oncologist and pathologist) and/or geriatric oncology clinic (the geriatrician, cardiologist, psychologist), and actively contributed to data collection and, at least in part, to study conceptualization. Some authors also were in charge of the prospective database update. The statistician helped with database cleaning and statistical analysis. The specific section has now been filled in.
We thank for your attention and interest,
Sincerely,
Benedetta Chiusole, MD
Department of Oncology, Oncology 1, Veneto Institute of Oncology IOV-IRCCS
Via Gattamelata, 64
35128 Padua, Italy
Email: benedetta.chiusole@iov.veneto.it
Reviewer 2 Report
1. In this retrospective cohort study, the authors looked at the predictive values of the CGA and the Onco-MPI on survival in patients >70 years with metastatic soft tissue sarcoma. They found that the CGA categories of frail, vulnerable, and fit, correlated with increasing survival, and that the onco-MPI categories of high and intermediate risk accurately predicted risk of death at 1 year.
2. There are a couple of major conceptual and statistical concerns that we found with the article that need to be addressed:
a. The authors mention their institution has been doing CGA’s on patients >75 years old with metastatic soft tissue sarcomas since 2009 – what is being done with the CGA information? If treatment decisions are based on it, the conclusion about CGA fitness accurately predicting survival is a circular argument.
b. What is the aim / hypothesis? The authors simply state “…we assessed the role of CGA, and CGA-based Onco- MPI, in patients with soft tissue sarcoma at our Institution.” The lack of aim makes it very difficult to interpret the results.
c. How did they assess if the model is overfit? It is concerning for being overfit given the small sample size and the large number of variables, which would increase the type II error and possibly the type I error.
d. Does the data violate Cox’s proportional hazard assumption? It should be stated how this was tested.
3. There are a number of minor issues that I will outline section by section:
a. Abstract:
i. What is the question you’re looking to answer? This needs to be included in the abstract, preferably just before the Methods section.
ii. Line 47-48: The “risk” predicted by the Onco-MPI should be stated (i.e. risk of death at 1 year)
iii. Line 53-54 – is this statistically significant? If so, include a p value. If not, remove the word significantly
iv. Line 63-64 – your study didn’t include younger patients so you cannot conclude this. You also don’t discuss it in the main article, so it should not be mentioned in the abstract.
b. Introduction: Overall, this is laid out nicely. A few comments:
i. Paragraph starting at line 81 – this could be expanded on – why do they have worse outcomes, why are they denied chemotherapy? Providing more specific, quantitative information would bolster the argument and rationale for doing this study.
ii. Lines 108-114 – explain the Onco-MPI more – what types of cancer has it been validated in? What ages? Why was it developed? Why did you choose it?
c. Methods:
i. Offer an explanation why age 70 was used as the cutoff.
ii. Based on Figure 1, the have excluded some patients. Include the exclusion criteria in this section.
d. Results:
i. Table 3: Include the Low Risk category in the Onco-MPI row (even though there were 0 patients), otherwise it looks like it was forgotten
ii. Table 4: Clarify whether you’re referring to an upfront dose reduction or dose reduction after toxicity
iii. Table 6: What is VGA? Is it supposed to be CGA? If not, it should be defined.
iv. Overall too many tables. Tables 8 and 9 could be combined
e. Discussion:
i. The results in table 4 were surprising and should be addressed in the discussion – why did fit patients have more G3/4 toxicities than vulnerable? Why did they have similar rates of dose reduction and second line chemotherapy? Moreover, this is not in line with what we would expect and so this should be discussed.
ii. Line 228-229 – this shouldn’t be a new paragraph
iii. Include the limitations of your study, including retrospective design, long accrual period, small sample size, etc.
Author Response
Dear Editor
Thank you for the opportunity to revise and resubmit our manuscript entitled “Role of geriatric assessment and oncological multidimensional prognostic index (onco-MPI) in older patients (age ≽70 years) with advanced soft tissue sarcoma in a real-world setting” to your journal.
We would like to thank the reviewers for their time and for providing us with helpful and constructive comments.
Here below, we have answered the referees’ questions point-by-point (answers in bold italics).
REVIEWER #2
- In this retrospective cohort study, the authors looked at the predictive values of the CGA and the Onco-MPI on survival in patients >70 years with metastatic soft tissue sarcoma. They found that the CGA categories of frail, vulnerable, and fit, correlated with increasing survival, and that the onco-MPI categories of high and intermediate risk accurately predicted risk of death at 1 year.
- There are a couple of major conceptual and statistical concerns that we found with the article that need to be addressed:
- The authors mention their institution has been doing CGA’s on patients >75 years old with metastatic soft tissue sarcomas since 2009 – what is being done with the CGA information? If treatment decisions are based on it, the conclusion about CGA fitness accurately predicting survival is a circular argument.
Thank you for the comment: indeed, CGA information served more as a guide for identifying possible vulnerabilities in older patients, but no formal procedure was in place to take decisions according to CGA. CGA output was used to send patients i.e. to nutritional consult, to geriatric consult, to rehabilitation consult if impairments were observed.
- What is the aim / hypothesis? The authors simply state “…we assessed the role of CGA, and CGA-based Onco- MPI, in patients with soft tissue sarcoma at our Institution.” The lack of aim makes it very difficult to interpret the results. Thank you for the comment, The aim of our study was to describe real-word management and outcomes of older patients with advanced stage soft tissue sarcoma (since no benchmark data are available for this population), along with assessing the ability of CGA and oncoMPI to predict survival in these patients. Text has been amended accordingly.
- How did they assess if the model is overfit? It is concerning for being overfit given the small sample size and the large number of variables, which would increase the type II error and possibly the type I error. We thank the reviewer for raising this point. Yet, we did not verify the model for overfitting, since number observations is low with a high variables number.
- Does the data violate Cox’s proportional hazard assumption? It should be stated how this was tested. Cox’s proportional hazard assumption have been graphically verified, and are respected for the variables considered in the model. This has now been specified in the text.
There are a number of minor issues that I will outline section by section:
- Abstract:
- What is the question you’re looking to answer? This needs to be included in the abstract, preferably just before the Methods section. The aim of our study was to describe real-word management and outcomes of older patients with advanced stage soft tissue sarcoma (since no benchmark data are available for this population), along with assessing the ability of CGA and oncoMPI to predict survival in these patients. The abstract has been amended accordingly.
- Line 47-48: The “risk” predicted by the Onco-MPI should be stated (i.e. risk of death at 1 year). Text updated.
- Line 53-54 – is this statistically significant? If so, include a p value. If not, remove the word significantly. Text updated.
- Line 63-64 – your study didn’t include younger patients so you cannot conclude this. You also don’t discuss it in the main article, so it should not be mentioned in the abstract. Text updated.
- Introduction: Overall, this is laid out nicely. A few comments:
- Paragraph starting at line 81 – this could be expanded on – why do they have worse outcomes, why are they denied chemotherapy? Providing more specific, quantitative information would bolster the argument and rationale for doing this study. Text updated.
- Lines 108-114 – explain the Onco-MPI more – what types of cancer has it been validated in? What ages? Why was it developed? Why did you choose it? Text updated.
- Methods:
- Offer an explanation why age 70 was used as the cutoff. The cut-off of 70 years is the one which has been increasingly used in the geriatric oncology literature in recent years (compared to the 65 years threshold which was used in the past). Such cut off does not have a clearcut explanation, yet it is believed to be more consistent with life expectancy in Italy (threshold for seniority in Italy) as well as the age in which age-related changes start to be more pronounced.
- Based on Figure 1, the have excluded some patients. Include the exclusion criteria in this section. Text updated.
- Results:
- Table 3: Include the Low Risk category in the Onco-MPI row (even though there were 0 patients), otherwise it looks like it was forgotten. Table updated.
- Table 4: Clarify whether you’re referring to an upfront dose reduction or dose reduction after toxicity. Table updated.
- Table 6: What is VGA? Is it supposed to be CGA? If not, it should be defined. Text corrected.
- Overall too many tables. Tables 8 and 9 could be combined. Thanks, this has now been done.
- Discussion:
- The results in table 4 were surprising and should be addressed in the discussion – why did fit patients have more G3/4 toxicities than vulnerable? Why did they have similar rates of dose reduction and second line chemotherapy? Moreover, this is not in line with what we would expect and so this should be discussed. Thank you for raising the point, the discussion has been implemented accordingly.
- Line 228-229 – this shouldn’t be a new paragraph. Text updated.
- Include the limitations of your study, including retrospective design, long accrual period, small sample size, etc. Thank you, text has been implemented accordingly.
We thank for your attention and interest,
Sincerely,
Benedetta Chiusole, MD
Department of Oncology, Oncology 1, Veneto Institute of Oncology IOV-IRCCS
Via Gattamelata, 64
35128 Padua, Italy
Email: benedetta.chiusole@iov.veneto.it
Round 2
Reviewer 2 Report
1. Conceptual –
a. Lack of aim – they did fix this. They wrote: “The aim of our study was to describe real-world management and outcomes of older patients with advanced stage soft tissue sarcoma (since no benchmark data are available for this population), along with assessing the ability of CGA and oncoMPI to predict survival in these patients. Text has been amended accordingly. These are 3 separate aims. Ambitious for a relatively small study. Given the study design and limitations, the aim should focus on management and survival outcomes (Table 4, Figures 2, 3, 4) and skip the big univariate and multivariate analyses tables (tables 5 and 6).
b. concern re: circular reasoning – the authors addressed this in that they said that the CGA results weren’t directly used to inform decision making. However, it is difficult to argue that the CGA findings would not have affected management decisions. While they addressed this in their response, it is not addressed in the paper and it needs to be added as a potential limitation.
2. Statistical -
a. With respect to an overfit model, their answer is problematic. Reducing the number of variables in the model and acknowledging the issue in limitations is important. We recommend focusing survival on the Onco-MPI rather than the CGA. More information is needed on the approach to model building. Looking at tables 5 and 6, several variables in the multivariable model could be eliminated, and wide confidence intervals in most others suggest the model is overfit due to lack of power. While it is acknowledged in the limitations, it is unclear why more steps are not taken in the modelling itself.
b. In addition, if one of the aims Is to look at the value of the Onco-MPI in predicting survival, the model should be formally compared to one without the Onco-MPI. This can be done using Harrell’s concordance statistic for time to event models.
Minor points:
· We wrote: The results in table 4 were surprising and should be addressed in the discussion – why did fit patients have more G3/4 toxicities than vulnerable? Why did they have similar rates of dose reduction and second line chemotherapy? Moreover, this is not in line with what we would expect and so this should be discussed. They wrote: Thank you for raising the point, the discussion has been implemented accordingly.
· With their clarification about how the CGA information was used in real life, this is an interesting point! Suggests that maybe the CGA-informed interventions basically put the vulnerable population on par with the fit population. This is worth mentioning in the discussion although we recognize the findings are preliminary.
Several of our prior suggestions have not been adequately addressed.
· We wrote: Table 3: Include the Low Risk category in the Onco-MPI row (even though there were 0 patients), otherwise it looks like it was forgotten. This has not been added to the table.
· We wrote: Table 4: Clarify whether you’re referring to an upfront dose reduction or dose reduction after toxicity. We still can’t tell whether it was upfront or after toxicity.
· We wrote: Line 47-48: The “risk” predicted by the Onco-MPI should be stated (i.e. risk of death at 1 year). It is still not clear in the text.
Other minor comments:
Table 1 label should be expanded to explain what the coefficients refer to or how they are supposed to be used.
Author Response
Dear Editor
thank you for the opportunity to revise and resubmit our manuscript entitled “Role of geriatric
assessment and oncological multidimensional prognostic index (onco-MPI) in older patients (age
≽70 years) with advanced soft tissue sarcoma in a real-world setting” to your journal.
We would like to thank the reviewers for their time and constructive comments. As for the further comments of reviewer #2, these required some further statistical discussion with our statitistician who was not available in the past couple of weeks. Please find below the answers to their comments (bold italics).
1. Conceptual
a. Lack of aim – they did fix this. They wrote: “The aim of our study was to describe real-world management and outcomes of older patients with advanced stage soft tissue sarcoma (since no benchmark data are available for this population), along with assessing the ability of CGA and oncoMPI to predict survival in these patients. Text has been amended accordingly. These are 3 separate aims. Ambitious for a relatively small study. Given the study design and limitations, the aim should focus on management and survival outcomes (Table 4, Figures 2, 3, 4) and skip the big univariate and multivariate analyses tables (tables 5 and 6).
Indeed there are some limitations due to the rare disease we are dealing with, and the consequent numbers which are of course low, and these are discussed. Moreover, geriatric oncology is a field in which high level evidence is often lacking due to the exclusion of “real” older patients from large randomized trials, which enroll commonly only fit older patients. Therefore, would like to keep the univariate and multivariate analyses on variables which may impact survival, since they might be hypothesis-generating and provide the basis for testing in multicentric studies. This has been already. This has already been done and published by our group in other more common cancers in older patients (i.e. Pierantoni F et al, Comprehensive geriatric assessment is an independent prognostic factor in older patients with metastatic renal cell cancer treated with first-line Sunitinib or Pazopanib: a single center experience. J Geriatr Oncol, 2021 Mar;12(2):290-297; Procaccio L et al, The oncological multidimensional prognostic index is a promising decision-making tool: A real-world analysis in older patients with metastatic colorectal cancer. Eur J Cancer 2022 Dec;177:112-119).
b. concern re: circular reasoning – the authors addressed this in that they said that the CGA results weren’t directly used to inform decision making. However, it is difficult to argue that the CGA findings would not have affected management decisions. While they addressed this in their response, it is not addressed in the paper and it needs to be added as a potential limitation.
Thank you for pointing out this. Such a limitation has now been more precisely discussed in the manuscript.
2. Statistical -
a. With respect to an overfit model, their answer is problematic. Reducing the number of variables in the model and acknowledging the issue in limitations is important. We recommend focusing survival on the Onco-MPI rather than the CGA. More information is needed on the approach to model building. Looking at tables 5 and 6, several variables in the multivariable model could be eliminated, and wide confidence intervals in most others suggest the model is overfit due to lack of power. While it is acknowledged in the limitations, it is unclear why more steps are not taken in the modelling itself.
We thank you for these very accurate observations. Indee, all the variables we have chosen are supposed to be predictive for survival either in soft tissue sarcoma or in older patients from literature data (i.e. for colorectal cancer in older patients, cfr Procaccio L et al, The oncological multidimensional prognostic index is a promising decision-making tool: A real-world analysis in older patients with metastatic colorectal cancer. Eur J Cancer 2022, 177:112-119; for soft tissue sarcoma in the elderly, cfr. Liu H-F et al. Clinical Features and Prognostic Factors in Elderly Ewing Sarcoma Patients Med Sci Monit. 2018; 24: 9370–9375; Tsuda Y et al. Impact of geriatric factors on surgical and prognostic outcomes in elderly patients with soft-tissue sarcoma. Jpn J Clin Oncol. 2017 May 1;47(5):422-429). Our model, which considers 11 variables, might therefore indeed be somewhat overfit, despite the number of events we registered. This lies the base to have larger studies dedicated to older patients with rare cancers such as sarcoma in order to provide further evidence for these hypotheses.
b. In addition, if one of the aims is to look at the value of the Onco-MPI in predicting survival, the model should be formally compared to one without the Onco-MPI. This can be done using Harrell’s concordance statistic for time to event models.
In our study we did not want to build a new predictive model, yet wanted to test in a homogeneous cohort of patients with rare cancers (soft tissue sarcoma) the reliability of oncoMPI in predicting overall survival by itself, and considered together with some other variables which are common survival predictors in oncology. OncoMPI by itself proved to be significantly associated with one-year survival in our cohort of older patients with metastatic soft tissue sarcoma (on the contrary, for example, this is not the case for older patients with glioblastoma, for whom only Balducci’s CGA was predictive of survival, cfr. Lombardi G et al. Validation of the Comprehensive Geriatric Assessment as a Predictor of Mortality in Elderly Glioblastoma Patients. Cancers (Basel) 2019 Oct 9;11(10):1509)
Minor points:
· We wrote: The results in table 4 were surprising and should be addressed in the discussion – why did fit patients have more G3/4 toxicities than vulnerable? Why did they have similar rates of dose reduction and second line chemotherapy? Moreover, this is not in line with what we would expect and so this should be discussed. “They wrote: Thank you for raising the point, the discussion has been implemented accordingly”. With their clarification about how the CGA information was used in real life, this is an interesting point! Suggests that maybe the CGA-informed interventions basically put the vulnerable population on par with the fit population. This is worth mentioning in the discussion although we recognize the findings are preliminary.
Thank you for your suggestion, we highlighted this point in the manuscript.
Several of our prior suggestions have not been adequately addressed.
· We wrote: Table 3: Include the Low Risk category in the Onco-MPI row (even though there were 0 patients), otherwise it looks like it was forgotten. This has not been added to the table.
We apologise for the typo, the correction has been now verified in the uploaded text.
· We wrote: Table 4: Clarify whether you’re referring to an upfront dose reduction or dose reduction after toxicity. We still can’t tell whether it was upfront or after toxicity.
Table 4 first column has been implemented specifying the upfront reduction.
· We wrote: Line 47-48: The “risk” predicted by the Onco-MPI should be stated (i.e. risk of death at 1 year). It is still not clear in the text.
We apologise, and amended the text which now reads “Onco-MPI is a CGA-based score which also takes into account tumour characteristics, classifying pts in three risk groups for death at 1 year”.
Other minor comments:
Table 1 label should be expanded to explain what the coefficients refer to or how they are supposed to be used.
Thank you, we suppose in this case you are reffering to Table 2 (oncoMPI). Coefficients are explained in the source document which is referred to in the bibliography #16. Statistical methods used to derive the oncoMPI are described in the method section of the original article. We do believe that re-explaining how onco-MPI was obtained when there is a clear explanation in the bibliography would make the text too heavy and would be beyond the scope of this manuscript, which simply applies a tool which has been previously published.
We truly thank you for the time, attention and interest for our manuscript.
Sincerely,
Benedetta Chiusole, MD

Round 3
Reviewer 2 Report
- Statement of aim - The aim is very clear in the abstract ("This is a single-Center retrospective study which aims at describing real-word management and outcomes of older pts with advanced stage STS and at assessing the ability of CGA and onco-MPI to predict survival in these pts."), however it is inconsistently stated throughout the paper, including the title. There should be more cohesiveness. For example, at the end of the introduction in the body of the article (lines 122-123), it should be clarified that the CGA and onco-MPI are being investigated to be predictive of what? Also the other two aims should also be stated here ( to describe real world management and outcomes of these patients.)
- The title should also reflect the aim(s) more clearly than the "role"
- Regarding the issue that the CGA has been used for years in their centre and interventions have been implemented based on the results - this is nicely discussed in the discussion, but should also be stated upfront in the article so that the reader has context. For example, lines 372-375 should be in the introduction, along with the fact that CGA has been performed in older patients at their institution for x number of years.
I still think that the study would be more focused and concise if the CGA were removed as part of the role and that the article focused on the predictive power of the onco-MPI, but based on their reply the authors prefer to keep this in the paper, which I understand.
Although I continue to disagree with some of the statistical choices, I do understand that the authors consulted a statistician and gave examples of other studies that have used similar, albeit methodologically suboptimal, approaches.
Author Response
Dear Editor
we do appreciate the time dedicated to the revision of our manuscript entitled “Role of geriatric
assessment and oncological multidimensional prognostic index (onco-MPI) in older patients (age
≽70 years) with advanced soft tissue sarcoma in a real-world setting”.
With regard to the comments raised by the reviewer, please find below our point to point answers (bold italics):
- Statement of aim - The aim is very clear in the abstract ("This is a single-Center retrospective study which aims at describing real-word management and outcomes of older pts with advanced stage STS and at assessing the ability of CGA and onco-MPI to predict survival in these pts."), however it is inconsistently stated throughout the paper, including the title. There should be more cohesiveness. For example, at the end of the introduction in the body of the article (lines 122-123), it should be clarified that the CGA and onco-MPI are being investigated to be predictive of what? Also the other two aims should also be stated here (to describe real world management and outcomes of these patients.)
- The title should also reflect the aim(s) more clearly than the "role"
According to this suggestion, we included a better explanation of the aim in the main text (introduction) and we agree to change the title, if Editor agrees too, to “Management and outcomes of older patients (age ≽70 years) with advanced soft tissue sarcoma and role of geriatric assess-ment and oncological multidimensional prognostic index (on-co-MPI) in a real-world setting”
- Regarding the issue that the CGA has been used for years in their centre and interventions have been implemented based on the results - this is nicely discussed in the discussion, but should also be stated upfront in the article so that the reader has context. For example, lines 372-375 should be in the introduction, along with the fact that CGA has been performed in older patients at their institution for x number of years.
Thank you for your suggestion, we moved these lines in the introduction.
We truly thank you for the time, attention and interest for our manuscript, all your comments have indeed improved our work.
Sincerely,
Benedetta Chiusole, MD
